# Comparative Analysis of the PIN Auxin Transporter Gene Family in Different Plant Species: A Focus on Structural and Expression Profiling of PINs in *Solanum tuberosum*

**DOI:** 10.3390/ijms20133270

**Published:** 2019-07-03

**Authors:** Chenghui Yang, Dongdong Wang, Chao Zhang, Nana Kong, Haoli Ma, Qin Chen

**Affiliations:** 1State Key Laboratory of Crop Stress Biology for Arid Areas, College of Agronomy, Northwest A&F University, Yangling 712100, China; 2School of Stomatology, Wuhan University, Wuhan 430072, China; 3College of Food Science and Engineering, Northwest A&F University, Yangling 712100, China

**Keywords:** auxin, PIN genes, potato, expression profiles, phytohormones

## Abstract

Plant growth and morphogenesis largely benefit from cell elongation and expansion and are normally regulated by environmental stimuli and endogenous hormones. Auxin, as one of the most significant plant growth regulators, controls various phases of plant growth and development. The PIN-FORMED (PIN) gene family of trans-membrane proteins considered as auxin efflux carriers plays a pivotal role in polar auxin transport and then mediates the growth of different plant tissues. In this study, the phylogenetic relationship and structural compositions of the PIN gene family in 19 plant species covering plant major lineages from algae to angiosperms were identified and analyzed by employing multiple bioinformatics methods. A total of 155 PIN genes were identified in these species and found that representative of the PIN gene family in algae came into existence and rapidly expanded in angiosperms (seed plants). The phylogenetic analysis indicated that the PIN proteins could be divided into 14 distinct clades, and the origin of PIN proteins could be traced back to the common ancestor of green algae. The structural analysis revealed that two putative types (canonical and noncanonical PINs) existed among the PIN proteins according to the length and the composition of the hydrophilic domain of the protein. The expression analysis of the PIN genes exhibited inordinate responsiveness to auxin (IAA) and ABA both in shoots and roots of *Solanum tuberosum*. While the majority of the *StPINs* were up-regulated in shoot and down-regulated in root by the two hormones. The majority of PIN genes had one or more putative auxin responses and ABA-inducible response elements in their promoter regions, respectively, implying that these phytohormones regulated the expression of *StPIN* genes. Our study emphasized the origin and expansion of the PIN gene family and aimed at providing useful insights for further structural and functional exploration of the PIN gene family in the future.

## 1. Introduction

The plant phytohormone, auxin, earliest discovered in plants for the function of promoting growth, played a critical role in multiple biological processes, such as the cambium cell division, branches of cell elongation, differential root, and stem growth achieved by increasing the plasticity of the cell wall. The distribution of auxin mostly concentrated in vigorously growing parts including coleoptile, apical meristem, cambium, fertilized ovary and young seeds in plants [1]. Auxin was mainly synthesized in expanding leaves along with shoot apical meristem and transported to various parts of the plant [2] by directional cell-to-cell transport called polar auxin transport (PAT) through active transport [3]. Three crucial plasma membrane (PM) auxin transporters were involved in the PAT system [4]. Of them, the plant specific PIN-FORMED (PIN) auxin efflux carriers were extensively studied polar-localized PM proteins, which had been investigated generally in *Arabidopsis thaliana*, *Oryza sativa* and *Solanum lycopersicum* [5,6,7,8]. Uneven distribution of auxin in plants gave the infflux and efflux carriers different roles in the unidirectional transport of IAA from the top of the plant morphology to the lower [9].

The polarized auxin efflux components were predicted by chemiosmotic hypothesis. The first putative efflux carrier component to be identified and isolated was *AtPIN1* in the model plant *Arabidopsis thaliana* [6]. The function of *AtPIN1* was characterized by the phenotype generated by the loss-of-function mutation in the gene. The morphological phenotype of *pin1* showed a stem nearly devoid of organs and generated naked, pin-shaped inflorescences, which could trace back to the shoot apical meristem for failing to form primordia [10]. *Atpin1* mutants could be phenocopied by correlating with a defect of cellular auxin efflux at the site of the inhibitor 1–naphthylphthalamic acid (NPA) action in polar auxin transport. Molecular cloning of the *PIN1* gene revealed that *PIN1* encoded a transmembrane protein with similarity to bacterial and eukaryotic carrier proteins [11]. The basal localization of *PIN1* in the plasma membranes provided for directional auxin flow in the vascular tissue of the stem, just as one would predict for the auxin efflux carrier postulated by the chemiosmotic hypothesis. At the same time, the homologous gene *AtPIN2* was identified as auxin efflux carrier complexes that performed a function in root gravitropic response on the basis of a strong root gravitropic phenotype of the loss-of-function mutant [6]. A number of transport assays showed that *PIN1* along with *PIN2* might impact the auxin efflux capacity of PIN proteins in different systems, which reflected on the relationship between PIN polarity and direction of auxin flow [12,13,14]. The analysis of expression and abundance of *PIN1* and *PIN2* in the respective mutants revealed that *PIN1* became ectopically induced in the *PIN2* expression domain in cortex and epidermis cells of *pin2*. Reciprocally, in pin1 mutants, *PIN2* was ectopically expressed in the endodermis and weakly in the stele [15]. Therefore, the auxin transport capacity in basipetal (shootward) direction, necessary for correct gravitropism, was not achieved. *PIN1* constructs with the insertion of the green fluorescent (GFP) coding sequence might interfere with a polarity determining signal [16]. Its localization was apically and able to rescue the agravitropism of *pin2* by mediating the auxin flux away from the root tip. Thus, the relation between PIN polar localization and auxin flow direction was proved by regulating the polarity of ectopically expressed *PIN1* and observing resultant auxin redistribution and tropic response [15].

In *Arabidopsis thaliana*, eight PIN proteins formed the PIN auxin efflux facilitator protein family [17]. These could be divided into two categories according to different classification bases. On the basis of localization in plant cells, PIN gene family members were localized in the plasma membrane (PM) including *PIN1-4* and *PIN7*, which were responsible for regulating intercellular auxin movement of plants [18]. Each member of the PM PINs had their respective functions. *PIN1*, *PIN3* and *PIN7* basally localized at stele cytomembranes in the primary root mediating the accumulation of auxin in the root tip [19]. *PIN2* was reported to be the pivotal PIN on regulating root meristem swelling for its special localization to the rootward and shootward plasma membrane of epidermis and cortex cells [20]. On the contrary, another type the ER PINs encompassed *PIN5* and *PIN8* with an endoplasmic reticulum localization in plant cells [21]. They were reported to balance cellular auxin homeostasis [22]. Based upon structures of eight PIN proteins described in Arabidopsis (*Arabidopsis thaliana*), two subfamilies were further divided in terms of the presence of central hydrophilic loop. *AtPIN1* to *AtPIN4* and *AtPIN7* formed the canonical PINs with long central hydrophilic loop separating two extremely conserved hydrophobic domains [23]. While the noncanonical PINs comprising *AtPIN5* and *AtPIN8* were subject to the absence of central hydrophilic loop [9]. The two types have been indicated to play important roles in plant developmental processes [24]. For instance, the canonical PINs were of great significance in gravitropic response of plant roots [9]. Besides, *AtPIN5* and *AtPIN8* might exist a certain regulation each other [25]. Thus, the noncanonical PINs might regulate the homeostasis of auxin in plant cells collaboratively [22]. *PIN6*, one of the PIN family members of auxin efflux carriers in Arabidopsis, was reported to differ from the defined ‘PM’ and ‘ER’ PINs or ‘canonical’ and ‘noncanonical’ PINs for it had localizations to the plasma membrane, as well as endoplasmic reticulum [26]. In addition, the hydrophobic loop of *PIN6* appeared to have Ser/Thr phosphorylation sites, which were usually a feature of the canonical PINs [27,28]. All in all, many studies shown that each PIN gene of *A. thaliana* played different physiological roles due to their specialized expression patterns in plant tissues and cells [20].

The PIN gene family had been extensively studied in various plant species. Up to now, no PIN representative had been identified from *chlamydomonas* [29]. Further evolutionary studies indicated that a PIN sequence identified from streptophyte algae belongs to another major lineage of green algae. In model moss *Physcomitrella patens*, four PIN proteins (*PINA-PIND*) had been characterized, including *PINA* and *PINC* localized in the ER and cytosol respectively [30]. In the evolution of the plantae, 6 PINs in *Selaginella moellendorffii* (a Lycophytes) [31], 1 PIN in *Picea abies* (a Gymnosperms) [32] had been identified. Until now, 10 PINs in potato (*Solanum tuberosum*) [9], 23 PINs in *Glycine max* [4], 12 PINs in *Zea mays* [31] had been reported in several researches, indicating uniform tissue-specific expression patterns. For example, *StPIN2* and *StPIN4*, which were highly homologous with *A. thaliana* PINs, exhibited a role for auxin in tuber development, especially stolon swelling [33]. *GmPIN9* was reported to be pivotal in auxin redistribution of soybean roots development in response to various abiotic stresses at the transcription level [4]. Interestingly, four *PIN1* maize orthologs showed highly expression specificity. *ZmPIN1a* and *ZmPIN1b* were ubiquitously expressed; *ZmPIN1c* was primarily detectable in the root cap tip and stem, while *ZmPIN1d* transcripts were predominantly localized in inflorescence meristem including tassels and ears [31]. In rice, four *OsPIN1* genes and one *OsPIN2* belonged to long PINs whereas shorter proteins such as three *OsPIN5* and one *OsPIN8* had been identified. In addition, *OsPIN9*, *OsPIN10a* and *OsPIN10b* were characterized as monocot-specific PINs [3]. 

Considering that the PIN gene family played a pivotal role in plant growth regulation, we described the evolutionary history of the PIN auxin efflux transporter gene family in 19 plant species through bioinformatics analysis covering algae, moss, lycophytes, ferns, gymnosperm, amborellales, and angiosperms. Then the gene characteristics, structures, and phylogenetic analysis were performed among major lineages of plants. Moreover, we carried out detailed structure analysis on 10 identified potato PIN genes. Meanwhile, we emphasized that they responded to hormonal treatments such as auxin (IAA) and abscisic acid (ABA) in different time intervals. The study aimed to reveal the evolution of the PIN gene family from lower to higher plants and focused on the identification of the potato PINs associated with hormone responses. Finally, we predicted the potential functional roles of potato PIN genes in various biological processes based on Gene ontology (GO) annotation. These results might be useful for further research especially into potato.

## 2. Results and Discussion

### 2.1. Identification of PIN Gene Family in Plants

In order to identify the PIN gene family in designated plant species, we used the Arabidopsis PIN proteins to query the genomes of 19 plant species encompassing lower and higher plants for screening all of potential PIN-like proteins in each species. The sequences acquired from local BLAST searches were verified by NCBI conserved domain database based on the presence of the auxin efflux carrier domain. Based upon above criteria, 155 putative PINs were identified from 19 plants across several major lineages such as algaes, moss, lycophytes, ferns, gymnosperm, amborellales, dicots and monocots. Our analysis confirmed that the number of PIN genes ranged from 1 to 23 among different plant species (Table 1). There was a maximum of 23 PIN genes in soybean, whereas only one PIN representative was detected in algae known as *Klebsormidiophyceae* (Streptophyte algae). Soybean had about three times PIN genes than Arabidopsis, while rice and maize had same number of the PIN genes which was only half as many as soybean. Among the other investigated species, presence of the PIN gene number varied from 4 to 16. In the NCBI-Genome, we found that *G. max*, *Z. mays* and *A. thaliana* contained 56,044, 40,557 and 27,416 gene locus, respectively. It was the gene loci number of soybeans that was short of two and three times than maize and Arabidopsis. This implied that the number of PINs was not proportional to the size of the genomes. According to the trend of evolution, representative of the PIN gene family in algae came into existence and rapidly expanded in angiosperms (seed plants), which was consistent with previous studies [6]. Thus, we assumed that plant species might suffer from some stresses to prompt richness or generate new functions to adapt to environmental changes in this gene family [34].

Plant evolution was accompanied by abundant gene duplication events which were pivotal in the expansion of gene families [35]. Gene duplication events were implemented via two common mechanisms: segmental duplication and tandem duplication [36]. In our study, gene duplication events were also analyzed against 19 investigated plant species by means of the Plant Genome Duplication Database (PGDD) [37]. There were hardly any duplication events selected in several lower plants as well as gymnosperm. Angiosperms, by contrast, more duplication events were identified. For example, of the 23 soybean PIN genes, 13 pairs involved in segmental duplication. There were 9 pairs out of 16 *P. trichocarpa* PINs and 6 pairs out of 12 *O. sativa* PINs, as well. Interestingly, both genes in a segmental duplication set were from same family. In addition, only two pairs of *StPIN* genes were confirmed as tandem duplicates based on defined criteria. Even though segmental duplication occurred more frequently in plants, both of the two duplication types largely facilitated the expansion of a number of gene families during plant evolution. 

### 2.2. Phylogenetic Analysis of the Putative PIN Proteins

A total of 154 putative full-length amino acid sequences of PIN genes from 18 plant species were aligned by ClustaXl (2.1) along with a partial EST-based PIN sequence of *Klebsormidiophyceae* (Streptophyte algae). In order to explore the phylogenetic relationship of PIN gene family in plants, a neighbor-joining tree was constructed by Mega7 based on the results of multiple sequence alignment. According to the topological structure of tree and sequence similarity, 14 distinct clades were classified and differentiated in color (Figure 1 and Appendix A). Previously, *Viaene T* et al. (2013) reported that PIN proteins originated in *Klebsormidiophyceae*, which was one of the first diverging lineages of streptophyte algae [29]. During land plant evolution, there were PIN representative genes that precisely appeared in bryophyte such as *Physcomitrella patens* in our study. Then in vascular plants, the appearance of PIN genes became prevalent. Additionally, substantial changes occurred in the number and structure of PIN genes since the divergence of monocots and dicots (Table 1). As illustrated in the phylogenetic tree, we employed *KfPIN* as the root on the basis of the previous study, although there were incomplete genome and transcriptomic data from it. *P. patens* (*PpPIND*) was closest to the origin of the gene tree together with several monocot PIN genes implying that the *P. patens* PIN genes were extremely likely to diverge from the earliest progenitor of PIN genes during evolution [38]. Further, it could be assumed that *PpPIND* might have been transferred from monocots horizontally [6]. In recent years, further investigations done by *Bennett* et al. (2014) confirmed later emergence of *Physcomitrella patens* [30]. Hence three *P. patens* PINs (*PINA-PINC*) counted in the phylogenetic analysis. It was interesting to find that the *P. patens* PINs clustered with PINs from *S. moellendorffii* in a single clade, nevertheless, it isolated from PINs of seed plants, which signified independent evolutionary relationships among each major lineage. As we know, the earliest and extensive research had been done on *A. thaliana* PIN gene family [6,17]. Following the evolutionary path of cladogram, the *A. thaliana* PINs generated several typical groups represented by five single branches *AtPIN1, AtPIN2, AtPIN5, AtPIN6, AtPIN8* and *AtPIN3, 4, 7* groups. Notably, three clades (*AtPIN5, AtPIN6* and *AtPIN8*) along with respective homologies emerged prior to that clades of *AtPIN1-4* and *AtPIN7*, which triggered our speculation that diverse natures of structures in PIN genes were largely able to determinate the mechanisms of evolution. Additionally, there were broadly conversed phylogenetic structures in dicot plants. For example, at least one or more genes homologous of all designated dicots except for *Utricularia gibba* showed similarities to each group of the *AtPINs* suggesting that these homologous genes clustered together might share the same or overlapping functions. Conversely, abscence of monocot-specific PIN homologous genes in *AtPIN3, 4, 7* group and *AtPIN6* clade compared with dicot PINs were observed. Moreover, three *OsPIN* proteins (*OsPIN9, OsPIN10a, OsPIN10b*) and *ZmPIN* proteins (*ZmPIN9, ZmPIN10a, ZmPIN10b*) had a conversed structure between each other. The generation of this exclusive type of genes potentially undertook functions that were specific to monocots and impacted on morphogical establishment [31]. Browsing through the overall evolution trend, the linage specific expansion via partial modification of the genome existed generally in paralogous gene families, supposing that this was a pattern to reinforce the adaption to environment stress [39,40]. 

### 2.3. Structural Analysis of PIN Proteins

The MEME (Multiple Em for Motif Elicitation) website was employed to reveal motif compositions of 66 PIN proteins belonging to nine representative species of seven evolutionary lineages from algae to angiosperm. In order to dissect structural comparisons between PIN proteins, 20 conversed motifs were identified. Throughout general structure analysis of all designated PIN proteins, two distinct types of motif compositions among these PIN proteins were observed. We also found that ten motifs were common in these PIN proteins that were located at the beginning (motif13, motif1, motif18, motif3 and motif5) and the end (motif4, motif7, motif6, motif12 and motif2) respectively. And the middle region comprised of several distinct motifs that were variant. As we know from the previous study, the structure of PIN proteins was predicted as two hydrophobic domains which were separated by a hydrophilic domain which was actually a central intracellular loop [6]. The structure of the loop was identified as the main functional domain of PIN proteins [25]. Hence, we focused on the comparative analysis of the loop structure within all PIN proteins and found that two groups of motifs (motif17, 14, 10 and motif 8, 16) and two individual motifs (motif11 and motif15) within the common modular loop were specific to the majority of PIN proteins. Therefore, we considered these proteins as canonical PIN proteins (including related variants) as defined by *Bennett T* et al. *(2014)* as well as structure compositions of these PIN proteins, which was represented by *PIN1-4, PIN6-7* in *A. thaliana* and *PINA-PINC* from *P. patens.* By contrast, the other PIN proteins was designated as noncanonical PIN proteins (also several related variants) for the absence of the modular loop, especially these two motifs and groups of motifs. For instance, *AtPIN5, AtPIN8* and *PpPIND* were typical. Moreover, to pinpoint the evolution of diversification of PIN proteins, we reconstructed a rectangular tree about 66 PIN proteins from nine species with two n/c proteins (*StPIN3* and *OsPIN1d*) excluded. The phylogeny revealed that the noncanonical PIN proteins were the ancestral type, which was identical to the previous findings [21,29]. However, further study demonstrated that the primary structure of noncanonical PIN proteins certainly diverged from the canonical type which was affirmed as one single ancestor of land plant [25]. Basically, the consequence of these divergent findings limited by few available data about algal PIN proteins, so that it was not clear which type of structure algal PIN proteins belong to. Even so, *Viaene T* et al. *(2013)* proposed that one of algal PIN proteins from *Klebsormidium* was probably a noncanonical type sequence. Subsequently, this type of PIN proteins appeared commonly in all land plants all except that there was no representative among Lycophyte (*Selaginella moellendorffii*), ferns and gymnosperms (*Picea abies*), suggesting that they might undergo provisional sampling from the transcriptome efforts [29]. At present, sequencing of partial non-seed plant representatives suffered incomplete sampling, so we could not exclude that further sampling might reveal the presence of short PINs in genomes of these species. The canonical type, by comparison, was present throughout all lineage since land plant diversification. 

Genes homologous to the Arabidopsis PIN sequences were present in genomes throughout the plant kingdom. These included representative in non-vascular plants such as physcomitrella; in vascular plants, PIN sequences appeared to be ubiquitous. General structural analysis of PIN proteins further suggested that two distinct types (noncanonical PIN proteins and canonical type) could be distinguished during green plant evolution (Figure 2). The noncanonical type of PIN protein originated in streptophyte algae following the PIN protein phylogeny and might appear to be present throughout all land plants just depending on further sampling from the transcriptome efforts in non-seed plant representatives such as ferns and gymnosperms. The canonical type appeared at the same point in evolutionary history as the origin of land plants and was present in all lineages that have evolved since then. However, we could not exclude that complete genome sequencing of streptophyte algae might reveal the presence of canonical PINs in genomes of these species. If further investigation did support to acquire more details for PIN protein phylogeny, the hydrophilic regions of the canonical type PINs could potentially mediate eukaryote-specific protein-protein interactions. Both before and since the divergence of monocot and dicot plants, there had been significant changes in the number and the structure of PINs. In dicot plants, the phylogenetic structure of the family was broadly conversed. In contrast, the monocots had a more divergent phylogenetic structure among several species, suggesting that these exclusively monocot PINs might undertake a function associated with physiological or morphogenetic processes which were specific to monocots such as development of the characteristic root system or phyllotaxy.

Similar to other plant PINs, potato (*S. tuberosum*) PIN proteins possessed a typical structure composed by a central hydrophilic domain within two highly conversed hydrophobic domains located at N- and C-termini respectively (Figure 3). At the N-terminus of the proteins, 7 out of the 10 *StPIN* proteins had 5 trans-membrane domains (Helix1-5). Exceptions were *StPIN7* and *StPIN8* with 4 trans-membrane domains for lack of helix3 and helix2, while *StPIN3* without any trans-membrane domain at its N-terminus. At C-terminus end, 3 to 5 trans-membrane domains were distributed with the exception of *StPIN8*, which presented only one trans-membrane domain (Helix7). Besides, the positions of these predicted helices were accurate and fell within refined model by a published study [25] (Appendix A). As for the central hydrophilic domain, four highly conversed domains HC1-HC4 were specific to most of *StPIN* proteins. Of course, these four domains were found in the majority of PIN proteins and proposed as a canonical PIN structure [25]. Previously, *Efstathios R* et al. (2013) divided *StPIN* proteins into different groups depending on their length of the hydrophilic region [9]. In our study, we concentrated on structural divergences between ten *StPIN* proteins and classified them into two types. Analysis of multiple sequence alignment among ten potato PIN proteins indicated that *StPIN1-4, StPIN6-7* and *StPIN9* exhibited all of HC1-HC4 domains orderly within the common central hydrophilic region and the consensus sequence of each domain scored at least 50% identity, thus identified as the canonical type PINs. In contrast, the remaining 3 *StPINs* (*StPIN5, StPIN8* and *StPIN10*) that belonged to noncanonical type lacked those four conversed domains including the central hydrophilic region. This noncanonical type PINs might have different functions from the canonical PIN proteins due to their divergent hydrophilic domains. One predicted function was that the noncanonical PINs could be carriers of auxin-like molecules such as indole butyric acid and auxin analog 2,4-D [22]. Hence, it could be postulated that novel modification in the hydrophilic domain of canonical PINs might lead to subfunctionilization by selective loss or repetition [25]. 

### 2.4. Gene Ontology Annotations of StPIN Proteins

Previously, there was an excellent research elaborated that a precise role for *StPIN2* and *StPIN4* proteins in redistributing auxin in the swelling stolon and developing tuber [9]. To better understand the biological processes related to *StPIN* proteins, we used the NCBI database to perform a GO annotation of potato PIN proteins. The results showed that the majority of *StPIN* proteins were participated in multiple biological processes such as signal transduction (10), cell communication (10), response to stimulus (10), homeostatic process (9) and so on. Majority of *StPINs* assumed a role on the regulation of these biological processes (Figure 4A). Besides, *StPIN* proteins also localized in multiple cellular components including organelle (9), cytoplasm (9), plasma membrane (9) even cell periphery (9) (Figure 4C). As it is known, the plant signaling molecule auxin mediates a variety of developmental processes by means of its differential distribution (gradients) within plant tissues [41]. Therefore, PIN-mediated directional trans-membrane and cell-to-cell transport activities were crucial for the steady-state level of auxin within cells. However, 9 out of 10 *StPIN* proteins might possess this activity according to the results of gene ontology annotation (Figure 4B). In Arabidopsis, the functions of most *AtPIN* proteins had already been characterized [20,21,22,26]. We found that high genetic homology between Arabidopsis thaliana and potato PIN genes based on the protein sequence alignment. Thus, we predicted that there might be similar functional roles or sub-functionalization between these homologous genes. Furthermore, we expect to deduce similar functional roles of *StPIN* proteins by this species-dependent development processes.

### 2.5. Expression Profiles of StPIN Family Genes in Response to Auxin and Abscisic acid Treatment

The growth regulator auxin was reported to mediate the adaption of plants to environment [42,43]. In Arabidopsis, auxin distribution mechanism relied on auxin feedback regulation of PIN gene expression at the transcriptional level [10,44,45]. Additionally, the phytohormone abscisic acid (ABA) played a pivotal role in plant physiological growth and development [46]. In order to find out the expression of auxin transporter encoding genes in responds to exogenous hormones, quantitative RT-PCR analysis was carried out with RNA isolated from shoot and root tissues of plantlets after IAA and ABA treatments with different time intervals. The results showed that each *StPIN* gene was inordinately responsive to IAA and ABA treatments both in shoots and roots in comparison with control. The expression of the *StPINs* were up-regulated in shoot and decreased in root due to the two hormones treatments. Further, the expression of some *StPINs (StPIN1-3, StPIN6* and *StPIN7)* went up to relatively high level after 6 h of IAA treatment. Similarly, the expression levels of these five PINs (*StPIN1, StPIN3 and StPIN5-7*) were also significantly increased after 12h of ABA treatment (Figure 5). Presently, research into PIN genes response to plant hormones in several species had been done extensively. Soybean PIN genes were reported to differentially express upon ABA and auxin treatments [4]. The effects of auxin and salicylic acid (SA) on cotton PIN genes showed that most *GhPINs* were responsive to the two hormones both in shoots and roots [24]. In addition, auxin transporter genes in maize all were identified to be IAA-responsive, which exerted different expression levels [47]. Moreover, the application of transgenic technology aimed to obtain PIN gene-specific overexpression or RNA interference transgenic plants might lead to a better understanding of their functions. 

To further explore the potential regulatory mechanism of ten potato PIN genes to phytohormone and how these genes regulated by auxin and abscisic acid, the PLACE web server [48] was employed to find their putative cis-elements in their 2000bp promoter regions. The putative phytohormones regulatory elements targeted as auxin response factor (ARF)-binding site (S000270), tissue-specific expression and auxin-inducible (S000273) and auxin-inducible (S000370); ABA responsive element (S000414) and ABA-inducible (S000174). Among the ten *StPIN* genes responded to hormone treatments, 10 and 6 were found to contain one or more putative auxin response elements and ABA-inducible response elements, respectively (Figure 5b,c), indicating that most of *StPIN* genes depended upon these elements to respond to IAA and ABA stimuli. These data would provide useful insights towards further functional understanding of this gene family in response to phytohormone stimuli. 

## 3. Materials and Methods

### 3.1. Sequence Retrieval and Identification of PIN Genes 

To identify potential PIN genes in 19 plant species (Charophyceae, Chlamydomonas reinhardtii, Klebsormidiophyceae, Physcomitrella patens, Selaginella moellendorffii, Cystopteris fragilis, Picea abies, Amborella trichopoda, Arabidopsis thaliana, Brassica oleracea, Glycine max, Populus trichocarpa, Solanum tuberosum, Gossypium raimondii, Utricularia gibba, Brachypodium distachyon, Oryza sativa, Phalaenopsis equestris, Zea mays), we first used eight A. thaliana PIN sequences as queries to perform basic local alignment search tool (BLAST) searches against the database downloaded from NCBI including protein sequences of selected 19 species with E-value ≤ 10^−10^. The PIN genes from A. thaliana were downloaded from The Arabidopsis Information Resources (TAIR) database. The NCBI conserved domain database (Available online: http://www.ncbi.nlm.nih.gov/cdd) and Pfam website (http://pfam.xfam.org/) were then used to confirm the presence of conserved domains of all resulting sequences.

Protein sequences of all identified putative PINs were stored for following analysis. Gene duplication was confirmed using criteria based on previous research [49]. See Additional file: Appendix A for accession numbers of all sequences used in this study.

### 3.2. Multiple Sequence Alignment and Phylogenetic Tree Construction 

The PIN protein sequences of 19 plant species were subjected to multiple sequence alignment carried out using the ClustalX2 (http://www.clustal.org/clustal2/) to identify the conserved domains and motifs. In order to further explore the evolutionary relationship of plant PINs, MEGA7 software (https://www.megasoftware.net/) was used to convert Aln files generated by ClustalX2 into MEGA format and then perform phylogenetic analysis of the PIN proteins. And the statistical parameters used to construct Neighbor-Joining (NJ) tree were as follows: 1000 bootstrap replications, poisson model, and pairwise deletions. 

### 3.3. Conversed Motifs Analysis of Several Typical Species 

The MEME web server (Available online: http://meme.sdsc.edu) [50] was used to identify finer motifs of PIN proteins under the parameters set as default values, expect for the maximum number of motifs is 20. Ten typical species from different lineages including algae (*Klebsormidiophyceae*), moss (*Physcomitrella patens*), lycophytes (*Selaginella moellendorffii*), ferns (*Cystopteris fragilis*), gymnosperm (*Picea abies*), amborellales (*Amborella trichopoda*), dicots (*Arabidopsis thaliana* and *Solanum tuberosum*) and monocots (*Oryza sativa* and *Zea mays*) were selected as candidates. Additionally, DNAMAN software (https://www.lynnon.com/pc/framepc.html) and TMHMM Server v.2.0 (http://www.cbs.dtu.dk/services/TMHMM/) were applied to identify the refined structure of *StPIN* proteins.

### 3.4. Plant Growth, Hormone Treatments and Tissue Collection

The potato variety ‘Desiree’ was used in the hormone treatments. The plantlets were cultivated in Murashige and Skoog (MS) medium with 2% sucrose, 0.8% agar and 0.05% MES (2-Morpholinoethanesulfonic Acid). Then 1M KOH or 2.5N HCl was used to adjust pH to 5.8. All plantlets were grown in a plant growth chamber for four weeks with a 16 h light (10000L×) and 8h dark (0L×) photoperiod at 22 ± 1 °C. For auxin treatment, the root of plantlets was soaked in liquid MS medium containing 10 μM IAA under the same growth conditions as before. The same methods were used for ABA treatments except 50 μM ABA was used instead of IAA. Shoots and roots were collected separately at 0 h (h), 1 h, 6 h and 12 h after treatments and tissues from four plants were pooled as one sample. For each treatment, samples were collected from the equivalent growth vigour of plantlets and three biological triplicates were assembled per treatment. All collected samples were frozen in liquid nitrogen immediately and kept at –80 °C for use.

### 3.5. RNA Extraction and Quantitative RT-PCR (qRT-PCR) Analysis

Total RNA was extracted from treated and control samples using high purity total RNA rapid extraction kit (BioTeKe, Beijing, China) and then Fast Super RT Kit cDNA with gDNase was used to perform reverse transcription according to the manufacturer’s instructions. Design of *StPIN* gene-specific primers for quantitative real time PCR (qRT-PCR) analysis were conducted using Primer Primer 5 software and synthesized in Sangon (Shanghai, China) (Appendix A). Quantitative RT-PCR (qRT-PCR) was carried out on the Bio-Rad CFX96 Real Time PCR System using 2 × RealStar Green Fast Mixture (GenStar, Beijing, China) according to manufacturer’s protocol and Elongation factor 1-a (*ef1-a*) as the internal reference gene to normalize the expression of *StPIN* genes [51]. PCR amplification was performed in a total volume of 10 μL reaction mixture containing 0.4 μL cDNA as template, 5 μL RealStar Green Fast Mixture (2×), 0.4 μL of each forward and reverse primer (10 μM) and RNase-free water up to 10 μL. The thermal cycling conditions were set as follows: initial activation 95 °C for 2 min, then 40 cycles of 95 °C for 15s, 60 °C for 15s and 72 °C for 30s. A melting curve was generated from 65 °C to 95 °C. 2^−∆∆Ct^ method [52] was employed to visualize the expression data about each treatment of each gene. All the expression analyses were based on three biological and two technical replicates. 

### 3.6. GO Annotation and Promoter Sequence Analysis of Potato PIN Genes 

2000-bp upstream sequences of potato PIN genes were obtained from Phytozome v12.1.6. and PLACE [48] was used to scan for putative phytohormone responsive elements in the promoter regions of *StPIN* genes.

The gene ontology (GO) analysis of potato PIN proteins were performed using the Blast2GO software (https://www.blast2go.com/). The full-length amino acid sequences of 10 *StPIN* proteins were uploaded to the program with the NCBI database chosen as the reference to analyze on three categories: biological process, molecular function and cellular component.

## 4. Conclusions

A comparative genomic analysis of the PIN gene family in 19 plant species was provided in this study. Identification and analysis of phylogenetic relationships of PIN gene family in plants among 155 PIN proteins totally were performed. There were 14 distinct clades classified and had a lineage specific expansion during plant evolution. Then, a structural analysis of PIN proteins belonging to ten representative species of major evolutionary lineages indicated that PIN genes were mainly considered two types, the canonical PIN proteins and the noncanonical PIN proteins, respectively. Furthermore, we focused on potato PIN genes to investigate their expression patterns in response to auxin and ABA treatment. The differential expression profiles of the *StPIN* genes suggested functional divergence between tissues during IAA and ABA treatments. Meanwhile, putative phytohormone responsive cis-elements analysis of potato PIN genes indicated that all potato PIN genes had one or more putative auxin response and ABA-inducible response elements in their promoter regions. These findings indicated that the expression of the *StPIN* genes might be regulated by these phytohormones. Additionally, GO analysis revealed potential functions for the *StPIN* proteins. Finally, this study aimed at providing useful insights for further structural and functional exploration of the PIN gene family in the future. 

## Figures and Tables

**Figure 1 ijms-20-03270-f001:**
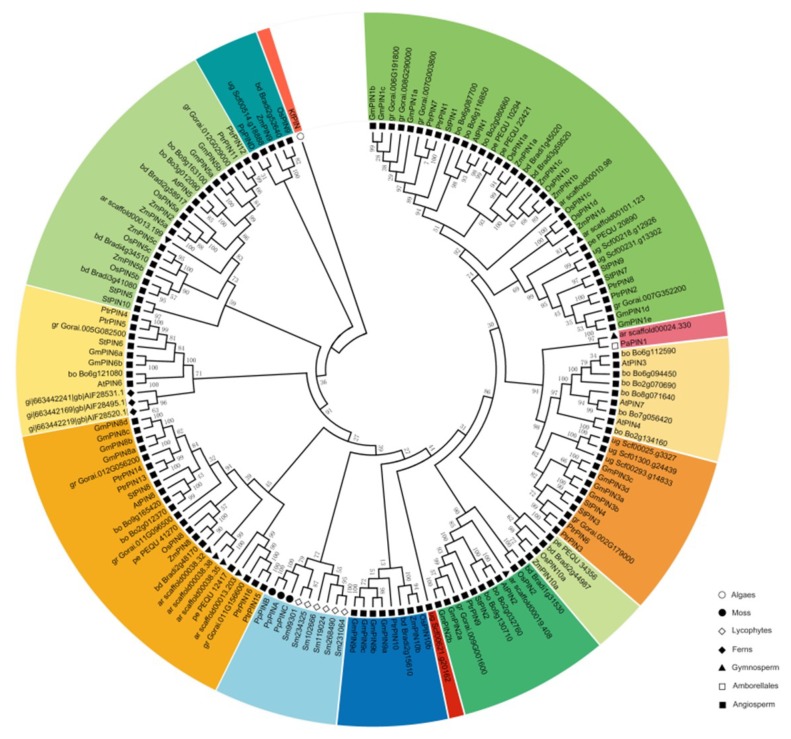
Phylogenetic analysis of PIN auxin transporter of 19 plant species. The putative full-length amino acid sequences of PIN genes were aligned by ClustaXI (2.1) with a partial EST-based PIN sequence of *Klebsormidiophyceae* (Streptophyte algae). Neighbor-joining tree was constructed by Mega7 with the statistical parameters using poisson model, and pairwise deletions. Accession numbers of the PIN proteins were listed in Additional file: Appendix A. The 155 PIN proteins from 19 plant species can be classified into 14 distinct clades and differentiated in color. Different lineages were distinguished by several geometric shapes with hollow or solid. The bootstrap values with 1000 replicates were placed on the nodes.

**Figure 2 ijms-20-03270-f002:**
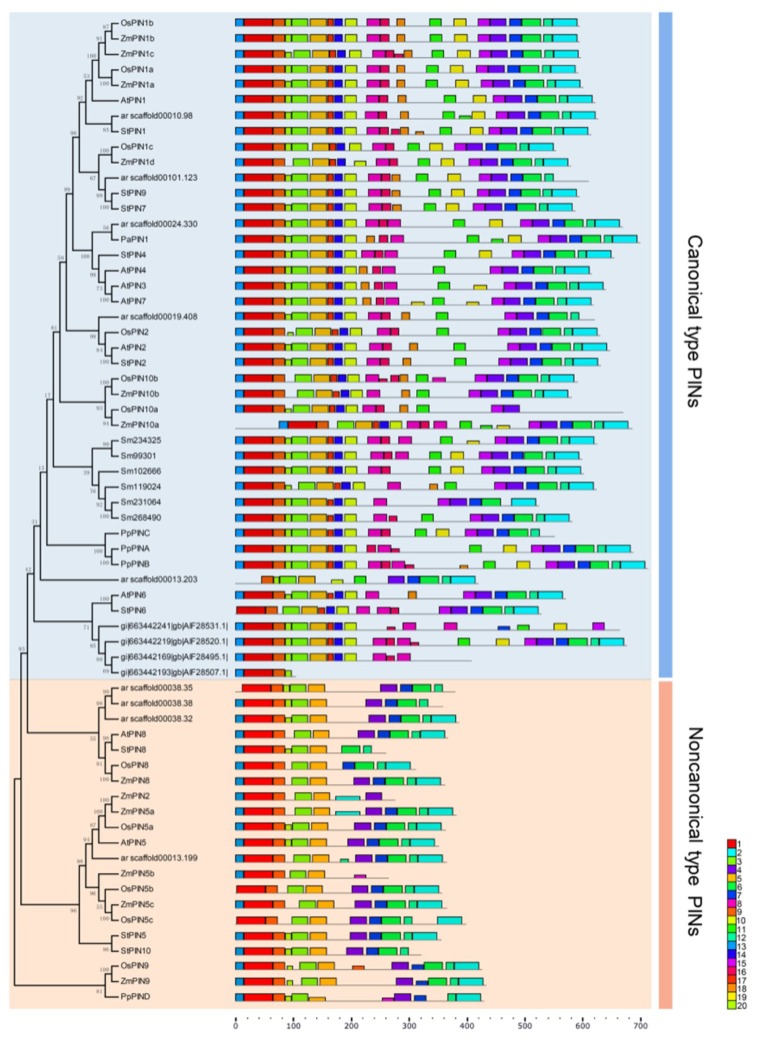
Motif compositions of the PIN genes in ten representative plant species. Phylogenetic tree of 66 PIN proteins from different lineages, including *K. flaccidium* (algae), *P. patens* (moss), *S. moellendorffii* (lycophytes), *C. fragilis* (ferns), *P. abies* (gymnosperm), *A. trichopoda* (amborellales), *A. thaliana* and *S. tuberosum* (dicots) and *O. sativa* and *Z. mays* (monocots). The unrooted Neighbor-joining phylogenetic tree was constructed with MEGA7 software. Predicted motifs of the PIN proteins were displayed by the MEME web server (Available online: http://meme.sdsc.edu) under the parameters set as default values, expect for the maximum number of motifs is 20. Twenty predicted motifs were shown by different colored boxes, and motif sizes were demonstrated by the scale on the right.

**Figure 3 ijms-20-03270-f003:**
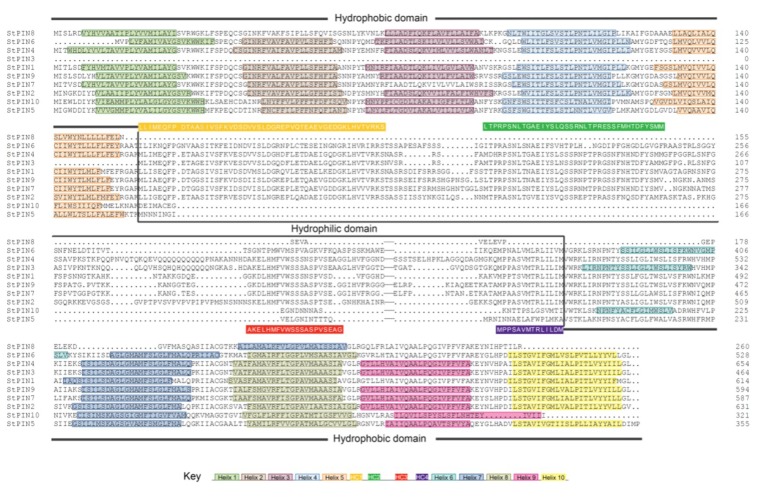
Trans-membrane and modular loop structure analysis of potato PIN proteins. Alignment results of *StPIN* proteins were conducted by DNAMAN software (https://www.lynnon.com/pc/framepc.html). Prediction of trans-membrane helices within two highly conversed hydrophobic domains located at N- and C-termini were performed by the TMHHM Server v.2.0 (http://www.cbs.dtu.dk/services/TMHMM/) and shown by the rectangle filling with distinct colors. The motifs that constituted the highly conserved canonical regions HC1-HC4 within the common central hydrophilic region were shown as yellow, green, red and purple rectangles, respectively. The positions of predicted helices in potato *StPIN* proteins were listed in additional files: Appendix A.

**Figure 4 ijms-20-03270-f004:**
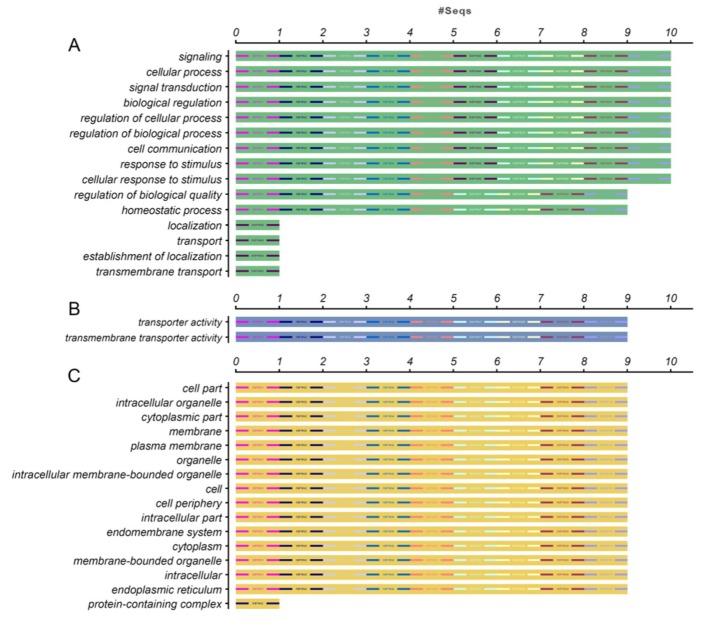
Gene ontology (GO) annotation of *StPIN* proteins. The gene ontology (GO) analysis of potato PIN proteins were performed using the Blast2GO software (https://www.blast2go.com/). The full-length amino acid sequences of 10 *StPIN* proteins were uploaded to the program with the NCBI database chosen as the reference. The annotation was performed on three categories, (**A**) biological process, (**B**) molecular function and (**C**) cellular component. The numbers on the abscissa indicate the number of predicted proteins.

**Figure 5 ijms-20-03270-f005:**
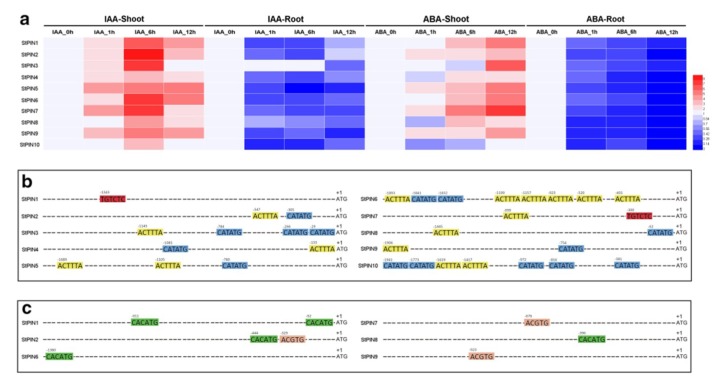
Quantitative RT-PCR analysis of *StPINs* from roots and shoots of potato plantlets treated with IAA and ABA. (**a**) Expression of *StPIN* genes after auxin and abscisic acid treatments. Data represented the means of three biological and two technical replicates in qRT-PCR analysis. The relative expression level was calculated by 2^-Δ ΔCT^ method comparing with internal standard (reference gene *ef1α*) and control, respectively, which was normalized to a value of 1 (Appendix A). (**b**) Analysis of *StPINs* with putative auxin responsive elements present in their promoter regions. The elements included auxin response factor (ARF)-binding site (S000270), tissue-specific expression and auxin-inducible (S000273) and auxin-inducible (S000370) marked with red, yellow and blue rectangle, respectively. (**c**) Analysis of *StPINs* with putative ABA responsive elements present in their promoter regions. The elements included ABA responsive element (S000414) and ABA-inducible (S000174) marked differently with pink and green rectangles.

**Table 1 ijms-20-03270-t001:** PIN-FORMED (PIN) genes identified in 19 sequenced plant genomes.

Lineage	Organism	Number of Predicted Genes (Loci) ^1^	Number of PIN Genes	Number of Long PIN Genes	Number of Short PIN Genes	Tandem Duplication (Pairs) ^2,3^	Segmental Duplication (Pairs) ^2,3^	References
Algaes	*Charophyceae*	300	0	0	0	0	0	This study
	*Chlamydomonas reinhardtii*	17,741	0	0	0	0	0	Viaene T., et al. [29]
	*Klebsormidiophyceae*	55,298	1	0	1	0	0	Viaene T., et al. [29]
Moss	*Physcomitrella patens*	32,926	4	3	1	0	0	Bennett, Tom A., et al. [25]
Lycophytes	*Selaginella moellendorffii*	22,273	6	6	0	0	1	Cristian, F., et al. [31]
Ferns	*Cystopteris fragilis*	7194	4	0	0	0	0	This study
Gymnosperm	*Picea abies*	28,354	1	1	0	1	0	Viaene T., et al. [29]
Amborellales	*Amborella trichopoda*	26,846	9	5	4	1	0	This study
Dicots	*Arabidopsis thaliana*	27,416	8	5	2	0	0	Paponov, I.A. et al. [17]
	*Brassica oleracea*	35,400	16	11	5	0	10	This study
	*Glycine max*	56,044	23	11	6	0	13	Wang, Y., et al. [4]
	*Populus trichocarpa*	41,335	16	7	6	0	9	Nicola, C. et al. [1]
	*Solanum tuberosum*	39,028	10	7	3	2	1	Roumeliotis, E. et al. [9]
	*Gossypium raimondii*	37,505	10	6	3	0	4	Zhang Y., et al. [3]
	*Utricularia gibba*	28,494	7	6	1	0	0	This study
Monocots	*Brachypodium distachyon*	34,310	10	5	5	0	5	Cristian, F., et al. [31]
	*Oryza sativa*	42,189	12	7	4	0	6	Wang, Y., et al. [4]
	*Phalaenopsis equestris*	31,384	6	4	2	0	0	This study
	*Zea mays*	40,557	12	7	4	0	5	Cristian, F., et al. [31]
Total			155	91	47			

^1^ The data were from NCBI (www.ncbi.nlm.nih.gov/genome/); ^2^ JGI (https://phytozome.jgi.doe.gov/pz/portal.html) contains hole information; ^3^ Related information obtained from PGDD (http://chibba.agtec.uga.edu/duplication/).

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
