# Peer review of "Comparative Analysis of the PIN Auxin Transporter Gene Family in Different Plant Species: A Focus on Structural and Expression Profiling of PINs in *Solanum tuberosum"

_ijms, 2019, doi:10.3390/ijms20133270_

Round 1
Reviewer 1 Report
The authors of manuscript “Comparative Analysis of the PIN Auxin Transporter Gene Family in 19 Plant Species During Evolution: a Focus on Structural and Expression profiling of Solanum tuberosum” described phylogenetic relationship and structural compositions of the PIN gene family in 19 plant species. The authors also predicted the functions of 10 PIN genes from potato. Overall, this manuscript provides a useful insight of PIN Auxin Transporter Gene Family. It can be accepted after addressing the following comments –
1. The authors need to check manuscript thoroughly for grammar and sentence structure.
2. What is the basis of choosing 19 plant families?
3. The 155 PIN proteins from 19 plant species are classified into 13 distinct clades or 14? Figure 1 shows 14 clades as differentiated by colors.
4. Page 13, lines 180-182 -The authors mention that “Thus we expect to excavate functional roles of StPIN proteins in species-dependent development processes based on high sequence similarity between AtPIN and StPIN genes”. What do they mean by this statement? As authors do only in silico studies to describe the roles of StPIN genes. In the future, do they intent to perform functional studies of these genes?
Reviewer 2 Report
Auxin (indole-3-acetic acid; IAA) is an important phytohormone that is involved in plant developmental processes, such as root development, embryo formation, vascular differentiation, etc. The plant specific PIN-FORMED (PIN) proteins are known as the most important auxin carriers that control the PAT (polar auxin transport) process, which results in the asymmetric distribution of auxin throughout the plant, consequently, control its growth and development. However, PIN protein families are extensively broad, and their functions and regulatory mechanisms vary among different plants. Moreover, there are still many PINs that remain to be discovered. Therefore, the authors aimed to do phylogenetic analysis using 155 putative PINs from 19 plants and showed the structural differences between canonical and noncanonical type PINs, which is very interesting and useful.
However, the paper has a major issue in its organization (please see the below) and each result section needs to expand the meaning of the data analysis, such as evolutionary importance of structural variation among PIN genes.
Also, the authors need to support the uniqueness of this study by providing, such as, what is the major difference in their finding compared to other studies. In addition, there is no take home message in the paper.
Overall, I have major issues with their Result and Discussion sections as detailed below, and the authors need to address the issues accordingly.
Major Issues:
1. Organizational issues:
Please change the order of sections 2.4 and 2.5: please place the section “2.4 Expression Profiles of StPIN Family Genes in Response to Auxin and Abscisic acid Treatment” after the section 2.5 Gene Ontology Annotations of StPIN Proteins.
2. the paper has a major issue in its organization (please see the below) and each result section needs to expand the meaning of the data analysis, such as evolutionary importance of structural variation among PIN genes.
3. In Figures 4 (b) and (c), the authors present Auxin responsive elements in the promoter regions of each St PIN gene. It seems that authors are trying to emphasize the correlation between the number of the Auxin responsive elements on each PIN promoter and the expression level of each gene after treating with Auxin and ABA in shoot and root. However, it is hard to see any correlation in this data and it is very confusing. Also, the authors need to add the biological meaning of this experiment: for instance, why each PIN gene is highly upregulated in shoot, and not in root when the potato plant is treated with Auxins. Is this simply due to a species-specific effect? Then, what does it mean under an evolutionary point of view, since the authors have done the comparative analysis among various plant species from lower to higher order plants? In addition, DNA elements are putative and there is no empirical data using DNA mutation. Therefore, the authors need to make it clear and use “putative Auxin responsive elements”.
4. Please provide the reason for picking the potato for studying the expression of each PINs.
5. What is the take away message for this study? I could not get it.
Minor issues:
1. Grammar issues.
2. The reference style is incorrect. Please see the reference 9.
3. The legends of the figures 3 (Key) & 4 are too small.
Reviewer 3 Report
Please See the extensive corrections made in the pdf file of the manuscript .

Reviewer 4 Report
Authors have done a great job in writing the manuscript, however, it lacks the significance and rationale of study in the introduction, discussion, and conclusion section. Therefore, I would recommend the author to add justification and hypothesis statement in the introduction. Additionally, there are couple grammatical typos and wrong usage of tense in several sentences which also need to be addressed in the revised version of the manuscript.
Reviewer 5 Report
The paper comprises two major sections. One is a descriptive view on the evolution of PIN gene family in plants, using essentially a set of very basic bioinformatic tools. The other is analysis of changes in the expression of potato PIN genes upon treatment with ABA and IBA. In the latter, the essential information on the methodology and results is presented in supplementary materials, which are not referred to in the text body, giving the impression that a significant portion of crucial information is lacking. Please, refer to the supplement in the text. Also, letters in figures are generally to small to be read, the figures must be enlarged and/or edited to provide readable information. The language of the paper is generally OK, but several grammar and working issues should be corrected.
Round 2
Reviewer 2 Report
The authors have modified the paper based on the reviewer's recommendation and the paper can now be accepted as is.